# Quadruplex Ligands in Cancer Therapy

**DOI:** 10.3390/cancers13133156

**Published:** 2021-06-24

**Authors:** Victoria Sanchez-Martin, Miguel Soriano, Jose Antonio Garcia-Salcedo

**Affiliations:** 1Centre for Genomics and Oncological Research, Pfizer-University of Granada-Andalusian Regional Government, PTS Granada, 18016 Granada, Spain; victoria.sanchez@genyo.es; 2Microbiology Unit, Biosanitary Research Institute IBS, University Hospital Virgen de las Nieves, 18014 Granada, Spain; 3Department of Biochemistry, Molecular Biology III and Immunology, University of Granada, 18016 Granada, Spain; 4Centre for Intensive Mediterranean Agrosystems and Agri-Food Biotechnology (CIAMBITAL), University of Almeria, 04001 Almeria, Spain

**Keywords:** cancer, G-quadruplexes, i-Motifs

## Abstract

**Simple Summary:**

Four-stranded nucleic acid secondary structures (quadruplexes) including DNA G-quadruplexes, RNA G-quadruplexes and i-Motifs display key regulatory functions in the human genome. Quadruplexes play an important role in telomere lengthening and the expression control of several cancer-related genes. In this context, quadruplex ligands are considered as potential strategies for anticancer drug discovery. Previous reviews are mainly focused on ligands targeting DNA G-quadruplexes, RNA G-quadruplexes and i-Motifs in a separate way, hindering a holistic study. The present review overcomes this limitation by providing a general overview of the recent research on ligands targeting the three different quadruplex structures in cancer.

**Abstract:**

Nucleic acids can adopt alternative secondary conformations including four-stranded structures known as quadruplexes. To date, quadruplexes have been demonstrated to exist both in human chromatin DNA and RNA. In particular, quadruplexes are found in guanine-rich sequences constituting G-quadruplexes, and in cytosine-rich sequences forming i-Motifs as a counterpart. Quadruplexes are associated with key biological processes ranging from transcription and translation of several oncogenes and tumor suppressors to telomeres maintenance and genome instability. In this context, quadruplexes have prompted investigations on their possible role in cancer biology and the evaluation of small-molecule ligands as potential therapeutic agents. This review aims to provide an updated close-up view of the literature on quadruplex ligands in cancer therapy, by grouping together ligands for DNA and RNA G-quadruplexes and DNA i-Motifs.

## 1. Introduction

Nucleic acids have considerable potential to fold into three-dimensional secondary structures based on particular sequence motifs [1]. Single-stranded guanine-rich DNA sequences can fold into stable intramolecular and intermolecular four-stranded G-quadruplexes (G4s) [2]. G4s arise from Hoogsteen hydrogen bonding of four guanines arranged within a planar quartet, which is further stabilized by interactions between the O-6 lone- pair electrons of each guanine and monovalent or divalent cations. Self-stacking of two or more G-quartets generates a G4 structure [3]. Further studies established that many RNA sequences featuring G-tracts can also fold into G4 structures, sometimes demonstrating increased thermodynamic stability and reduced steric hindrance [4]. Therefore, G4s are found both in DNA and RNA. In addition, single-stranded cytosine-rich sequences can form hemiprotonated cytosine–cytosine base pairs (C–C+), adopting a structure called an i-Motif [5]. This structure is less known than other DNA structures, in part due to its limited stability. Initially, i-Motifs were considered to mainly occur under acidic conditions, but an increasing number of studies demonstrate that these sequences are stable structures even at neutral pH [6]. Although quadruplexes are related to each other in terms of primary sequence, they in fact comprise a diverse family of structures that can fold into different topologies including parallel, antiparallel, and hybrid structures for G4s and R-forms or S-forms for i-Motifs (Figure 1). Each topology is dictated by the pattern of strand polarities and the orientation of interconnecting loops [7]. The topology of RNA G4s is more limited to the parallel conformation due to the steric hindrance imposed by the presence of a 2′-hydroxyl group in the ribose sugar. Nevertheless, according to a recent study, RNA G4s can also adopt different conformations other than the parallel [8]. However, the extent to which distinct topologies can influence quadruplex formation and function in cells is still unknown.

Experiments using computational, chemical, molecular, and cell biology methods have demonstrated that quadruplexes are very numerous in the human genome. The use of computational algorithms to search for the consensus sequence of quadruplexes helped to identify quadruplexes and reveal their enrichment [9]. Recent advances even accommodate certain structural variants, higher-order assemblies and account for flanking sequence effects [10]. Biophysical studies including circular dichroism and ultraviolet melting on different oligonucleotides, were the first to establish that many DNA and RNA sequences can fold into quadruplexes [11]. More sophisticated techniques, such as X-ray crystallography and nuclear magnetic resonance (NMR) spectroscopy, have been used to obtain quadruplex structures at atomic resolution. Interestingly, NMR spectroscopy simultaneously allows for the structure to be determined in physiological solutions, kinetics and dynamics studies, and molecular interaction investigations [12]. In addition, quadruplexes have been identified as cellular features. Given that G4s in DNA or RNA can stall a DNA polymerase or a reverse transcriptase respectively, the comparison of pause sites in G4-stabilizing and non-stabilizing conditions enables the detection of G4s in vitro. In particular, direct G4 sequencing in purified single-stranded human DNA identified more than 700,000 DNA G4s [13]. Furthermore, direct RNA G4 sequencing on poly(A)-enriched RNAs mapped G4 structures in more than 3000 human mRNAs [14]. Although sequencing of i-Motifs has not yet been achieved, C-rich complementary strands always accompany G-rich sequences in the genomic DNA. Therefore, the number of i-Motifs is expected to be similar to that of DNA G4s. Quadruplexes have been also identified using chemical methods by exploiting the different reactivities of nucleobases following the formation of quadruplex structures. For instance, in KMnO4–S1 nuclease footprinting, only single-stranded DNA is digested by S1 nuclease and subsequent computational analyses infer the formation of DNA G4s based on the nuclease footprints [15]. Alternatively, the location of G4s can be deduced because the Hoogsteen hydrogen interactions between the guanines in guanine tetrads provide relative protection from methylation as a result of dimethyl sulfate (DMS) and subsequent cleavage by piperidine [16]. Another method is the selective 2′-hydroxyl acylation analyzed by primer extension (SHAPE) which utilizes differences in the acylation kinetics of RNA 2′-hydroxyl groups and the ability of these groups to stall reverse transcription [17]. However, to date, these chemical mapping techniques have no detected RNA G4s in eukaryotic cells [18]. Furthermore, quadruplexes have been visualized by immunofluorescence in cells using specific antibodies. The scFv antibody BG4 revealed both DNA [19] and RNA [20] G4s in human cells. In accordance, the antibody fragment iMab enabled the imaging of i-Motifs in the nuclei of human cells and revealed that their formation was also cell-cycle dependent [21]. In general, quadruplex sequences are non-randomly distributed but are mainly clustered in pivotal genomic regions, such as DNA replication origins, telomeres, gene promoters, and untranslated regions (UTRs). There, quadruplexes either act as physical obstacles, which must be overcome, or as facilitators of normal cellular functions. Interestingly, quadruplexes are primarily found in active promoters associated with elevated transcription. However, the folding of quadruplexes in promoters is favored by an accessible chromatin state but does not necessarily require active transcription. In fact, quadruplexes act as genomic features that enable the recruitment of Polymerase II to promoters [22]. Altogether this evidence suggests that quadruplexes are involved in the regulation of different biological pathways such as replication, transcription, translation, and genome instability [23] (Figure 2A). 

In this regard, quadruplexes display key cancer-related functions [24] (Figure 2B). Quadruplexes are linked to the control of the expression of several oncogenes and tumor suppressors, both at the transcriptional and translational levels. In addition, quadruplexes participate in lengthening telomeres and induce genome instability, processes which are frequently altered in cancer in order to sustain limitless replication [25]. In fact, quadruplex-containing genes are involved in all of the six hallmarks of cancer: sustaining proliferative signaling, evading growth suppressors, resisting cell death, enabling replicative immortality, inducing angiogenesis and activating invasion and metastasis [26]. The quadruplex-containing genes related to cancer were comprehensively reviewed in our previous article [27]. In particular, DNA G4s are found in the repetitive (TTAGGG)_n_ telomeric sequence [28], which influences the binding of human telomerase reverse transcriptase hTERT, which is itself responsible for telomere lengthening. Interestingly, the expression level of the *hTERT* gene is also controlled by DNA G4s [29]. Apart from their association with telomeric function, numerous DNA G4s are harbored in the promoter regions of oncogenes. Within this group, DNA G4s are found in several oncogenic transcription factors and transducers, oncogenic growth factors and respective growth factor receptors. Tumor suppressor genes also contain DNA G4 structures in their promoters [24]. Regarding RNA G4s, they are found in telomeric repeat-containing RNA (*TERRA*) [30], and in the transcript of *hTERT* [31], controlling telomeric functions. Moreover, UTRs of certain cancer-related genes possess G4 sequences capable of mediating translation inhibition or activation and interfere with microRNAs (miRNAs) binding. RNA G4s are more abundant in the 5′UTR regions of oncogenes but they are also present in 3′UTRs [32]. Furthermore, RNA G4 structures in introns affect the splicing and expression patterns of several genes [33]. RNA G4s are found in tumorigenesis-involved noncoding RNAs, including both long noncoding RNAs (lncRNAs) and miRNAs. In particular, a G4 folded in a mature miRNA can prevent the miRNA from binding to its target, whilst a G4 within the pre-miRNA either positively or negatively influences its processing and biogenesis [34]. In the same way, C-tracts forming i-Motifs are found in telomeres [35] and extratelomeric regions, including the promoter of several oncogenes and tumor suppressors [36].

In this context, quadruplex ligands have been developed and tested as tools with which to study the complexity of quadruplexes-mediated mechanisms in carcinogenesis. Therefore, thanks to its stabilization properties, the small molecule, pyridostatin, enabled DNA G4 imaging in the nuclei of cells using the G4-recognizing antibody, BG4 [19]. The same occurred for carboxypyridostatin, a stabilizing ligand targeting RNA G4s within a cellular context [20]. Aside from being solely considered tools for improving visualization, quadruplex ligands have emerged as potential strategies for anticancer drug discovery [37]. 

This study aimed to concisely review the most recent advances in quadruplex targeting in antitumoral therapy. Given that the field of quadruplexes is continuously developing, we cover the current state of the art of quadruplex ligands in cancer research. Finally, we highlight the critical questions that remain to be addressed in the promising “era of quadruplexes”. Moreover, previous reviews separately focus on DNA G4, RNA G4 and i-Motif ligands in cancer. To the best of our knowledge, the present review is the first to bring together these quadruplex ligands, all of which are relevant for cancer therapeutics. 

## 2. Quadruplex Ligands

Enormous efforts are being made to target quadruplexes as a therapeutic approach given their profound implication in carcinogenesis [37]. Ligands are chemical compounds that specifically bind to and stabilize the structure of quadruplexes. Without this mechanism, quadruplexes would unfold immediately after their formation in the cell as a result of helicases [38]. Quadruplexes provide recognition sites for ligands since different quadruplex structures adopt specific conformations (Figure 3). Binders generally have an aromatic surface for π–π stacking interactions with quadruplexes, a positive charge or basic groups to selectively bind to the loops or grooves of the quadruplex, and a steric bulk to prevent intercalation with double-stranded DNA [39]. 

To date, an arsenal of around 1000 small molecules that target quadruplexes has been reported. In the section below, quadruplex ligands with an anticancer effect are summarized. Furthermore, the respective timeline is shown in Figure 4. To note, this timeline shows the year in which the compounds were first described as quadruplex interactive ligands (not when they were firstly synthesized or discovered). The majority of ligands have emerged in recent years. As a result of the extensive panel of existing ligands and the huge variety of in vitro experiments described in the literature, only the most relevant concluding remarks are included.

### 2.1. DNA G4 Ligands

Herein we review the plethora of existing small molecules targeting DNA G4s, from classical ligands to the most recently discovered and selective binders, through the main chemo-families (Table 1).

#### 2.1.1. Classical G4 Ligands

Disubstituted amidoanthraquinones were the first G4 ligands to be reported. These interactive agents tightly bind to telomeric G4s and block the telomerase RNA activity by disrupting the base-pairing between G-overhang and the enzyme [61]. One year later, the cationic porphyrin TMPyP4 emerged. TMPyP4 is traditionally known for inducing telomerase inhibition upon binding to telomeric G4s [114]. Broader targets include G4s in oncogenes such as: *c-MYC* transcription factor [111], *VEGF* vascular endothelial growth factor [115], *PDGFa* platelet derived growth factor [113], *k-RAS,* the GTP-binding protein involved in transduction [112], and *BCL2* mitochondrial protein, which regulates apoptosis [110]. However, a major hurdle in the development of TMPyP4 as a G4 ligand is its ability to also bind to duplex DNA [123] and triplex DNA [124]. The naturally occurring antibiotic alkaloid derived from Chinese herbal medicine, berberine, along with its derivatives, bind to DNA G4s and inhibit telomere elongation [47,56]. Interestingly, epiberberin was the first molecule to be reported that specifically recognized the hybrid-2 form, one of the telomeric G4 conformations, and was capable of converting other telomeric G4 forms to hybrid-2 [125]. In addition, berberine can form complexes with G4 *k-RAS* promoter in a molecular ratio of 1:1 [46]. Moreover, berberine can be encapsulated by a modified b-cyclodextrin, exhibiting a significantly stronger binding to the G4 of the tyrosine kinase receptor *c-KIT* [45]. Another natural product, telomestatin, isolated from *Streptomyces anulatus*, was considered a potent telomerase inhibitor due to its ability to facilitate the formation of G4 structures, and thereby sequestering the single-stranded (TTAGGG)_n_ telomeric primer molecules required for telomerase activity [108]. More recently, the (S)-telomestatin stereoisomer was shown to be a potent telomeric G4 binder and telomerase inhibitor [126]. Apart from G4s in telomeres, the transition of duplex to a non-B DNA conformation within the promoter region of the transcription factor *c-MYB* is induced by telomestatin [107]. The synthetic pentacyclic acridine RHPS4 also acts as a potent inhibitor of human telomerase on the submicromolar range through the stabilization of G4s formed by telomeric DNA in vitro [101]. Since its therapeutic potency is compromised by off-target effects on cardiovascular physiology, novel RHPS4-derivative ligands with improved toxicological profiles and telomere-targeting activities have been developed [127]. Bisquinolinium compounds, such as synthetic Phen-DC3, exhibit an exceptional affinity and selectivity for DNA G4s in human telomeric repeats over DNA duplexes [92] and form complexes with intramolecular G4s derived from the *c-MYC* promoter [91]. Moreover, pyridostatin is a synthetic small molecule that binds and stabilizes telomeric G4s leading to an alteration of shelterin complex integrity and the activation of DNA damage response [128]. Nevertheless, further studies indicate that, at low concentrations, pyridostatin predominantly interacts with non telomeric DNA loci, including several oncogenes, before targeting telomeres at higher doses [129]. As a result of its pan-quadruplex binding ability, pyridostatin has been used to broadly promote G4 formation in high-throughput sequencing of DNA G4 structures [130]. The 3,6,9-trisubstituted acridine compound, BRACO-19, was the first rationally designed ligand whose biological activity was partially restricted to G4-telomere targeting and interference with the capping and catalytic functions of telomerase [48]. 

Subsequently, some of these synthetic ligands have entered clinical trials (Figure 5). CX-3543, also named quarfloxin, is a fluoroquinolone that was originally designed to target the G4 found in the *c-MYC* promoter [131]. Further studies demonstrate that CX-3543 also interacts with a G4 found in ribosomal DNA and disrupts the binding between these G4s and nucleolin complexes in the nucleolus, thereby inhibiting ribosome biogenesis [132]. Although CX-3543 passed phase II trials as a candidate therapeutic agent against several tumors, phase III trials were not completed because of its high binding to albumin. Another fluoroquinolone, CX-5461, was found to selectively inhibit ribosomal RNA synthesis by reducing the binding affinity of the SL1 pre-initiation complex and RNA polymerase I complex to the ribosomal DNA promoter [133]. Very recently, it was shown that SL1 recruitment to ribosomal DNA is performed in a G4-dependent manner and CX-5461 traps such G4 structures, interfering with SL1 DNA binding activity [134]. Similar to CX-3543, CX-5461 selectively binds and stabilizes a broad spectrum of G4 structures, including those harbored in *c-MYC*, *c-KIT*, and telomeres [55]. Notably, CX-5461 is currently in phase I clinical trials for patients with *BRCA1/2* deficient tumors, constituting the most advanced G4 ligand in the clinics at the moment. 

#### 2.1.2. Chemo-Families of G4 Ligands

Many G4 ligands have characteristic cores that can be chemically modified, rendering various analogues whose therapeutic activity in cancer is being investigated (Figure 6). During the last decade, extensive research efforts identified naphthalene diimides (NDIs) as favored chemotypes for G4 binding due to their high target affinity and potential for chemical variability. NDIs were originally reported to bind to telomeric G4s resulting in telomerase inhibition [83], although they display a lower specificity for telomere targeting as compared to RHPS4 [135]. However, after easily tunable synthesis, NDIs were able to target several oncogene promoters, for example *c-KIT* [80,136] and *BCL2* [79]. A recent study further demonstrated that NDI derivatives stabilize the G4 formed in the promoter regions of *c-KIT* and *BCL2* leading to the suppression of their respective protein expression and thus, interfering with their oncogenic signaling pathways [137]. In addition, another NDI derivative binds to G4s in *MDM2* oncogene, which is a master regulator of *TP53*, reducing MDM2 transcription. Such an approach could be used to defeat all tumors in which the restoration of wild-type TP53 is sought [81]. Recently, our research group unveiled G4s in ribosomal DNA as new targets for NDIs [82]. Various NDIs, such as MM41 [138] and CM03 [139] demonstrated promising results in cancer therapeutics in vivo. Importantly, CM03 has potential for clinical use in the treatment of drug-resistant cancer. In particular, gemcitabine-resistant cancer cell lines demonstrate sensitivity to CM03 in the nanomolar range because the pattern of pathways downregulated by CM03 is largely unaffected in the gemcitabine-resistant line [140]. On the other hand, phenanthroline derivatives, originally used as duplex DNA-intercalating agents, have been modified to behave as telomeric G4-stabilizing ligands [90] and to induce the formation of a telomeric G4 antiparallel structure [60]. Aside from telomeres, phenanthrolines exhibit high selectivity for G4 DNA present in the promoter of *c-KIT* and *c-MYC* [89]. Inspired by the structure of the natural G4 ligand telomestatin, new insights were utilized in the synthesis of oxazole telomestatin derivatives (OTD), including L2H2-6OTD [86] and L1H1-7OTD [87], which are powerful scaffolds for stabilizing telomeric antiparallel G4s. Benzo[a]phenoxazine (BPO) derivatives have also been identified as new G4 binding molecules with a higher affinity for *c-KIT*, inhibiting its transcriptional expression [42]. Another benzo[a]phenoxazine, cresyl violet, demonstrates stronger binding activity to the (3+1) hybrid G4 structure formed by the human telomeric sequence as compared to the antiparallel one [43]. Quinazoline and its derivatives, which are building blocks for natural alkaloids, were reported to bind more selectively to *c-KIT* G4s than to duplex DNA [98]. However, in a more recent study, these compounds efficiently stabilized G4s of different topologies with a very strong preference for the well-characterized parallel *c-MYC* G4 [99]. Accordingly, the derivative NSC194598 was found to be a G4 interactive agent that interferes with transcriptional activation of the mutated *RET* gene in cancer, encoding a receptor-type tyrosine kinase [100]. Other naturally occurring alkaloids, indoloquinoline (IQ) derivatives, target G4s at the telomeres and oncogenic promoters, suggesting an inter-G4 selectivity trend: telomeric ≈ *k-RAS* > *c-KIT* [69]. In fact, these compounds are considered to be potent and selective *k-RAS* G4 stabilizers, that preferentially target the mutant form of *k-RAS* [141]. The rational selection of ligand side chains is crucial for enhancing the affinity or selectivity of IQ derivatives [142]. In this regard, methylated IQ derivatives exhibit a high affinity for the parallel *c-MYC* G4 in the submicromolar range, with a topology-selective binding and an excellent discrimination against the antiparallel telomeric G4 [70]. Moreover, another quindoline derivative, SYUIQ-05, interacts with the *VEGF* promoter, stabilizing its G4, thus downregulating its transcription and exhibiting a strong antiangiogenic activity [71]. Within the family of anthraquinone derivatives, mitoxantrone, is currently used in clinics for cancer treatment in combination with other drugs. Interestingly, recent studies revealed a new role for mitoxantrone in G4-dependent telomerase inhibition [76]. Additionally, mitoxantrone was shown to have potential in stabilizing G4s in the gene promoter of *WT1*, a zinc-finger transcription factor, downregulating its transcription [77].

#### 2.1.3. Selective G4 Ligands

Over the last decade, attention has been focused on the development of G4 binders with selective recognition. Firstly, we would like to note that the majority of ligands have not been evaluated in large panels of quadruplexes or across the human genome. In this regard, many of the ligands described in this section are rather promiscuous. Selectivity is further discussed in the Discussion section.

Initially, research studies were focused on small molecules with G4-depending telomerase inhibitory activity in telomerase-positive tumor cells. Spermine-derivatized tentacle porphyrins, including TCPPSpm4 and its Zn (II) derivative (ZnTCPPSpm4), stabilize telomeric G4 with high stoichiometry and point-to-end stacking or porphyrin self-association as major binding modes [119]. In addition, 4,5-diazafluorenes [121] and stiff-stilbenes [105] emerged as new G4-binding chemotypes displaying selectivity for telomeric G4. The natural compound quercetin even interacts with telomeric G4 in acidic conditions [97]. Novel derivatives of *Schizocommunin*, an alkaloid from a fungal source, have been also designed for telomeric G4 targeting [104]. Anticancer drugs, such as epirubicin [64] and adriamycin [40] bind as monomers to telomeric G4 with a high affinity. Interestingly, various telomeric G4 ligands have been shown to discriminate between dimeric and monomeric forms of G4s. For instance, the cationic porphyrin derivative m-TMPipEOPP [78], the triaryl-substituted imidazole derivative IZNP-1 [74] and aryl-substituted imidazole DIZ-3 [62] exhibited a highly specific multimeric interaction. Moreover, it is increasingly clear that chiral complexes present a significant enantioselectivity [143]. Metallo-supramolecular Ni-P complexes display selectivity in stabilizing monomeric G4s [85], whilst Ni-M complexes bind to higher-order G4s [84]. Λ-enantiomer of ruthenium complexes can also selectively stabilize human telomeric G4 [122]. Other Ruthenium (II) Schiff base complexes exhibit telomeric G4 targeting and photo-induce cancer cell death with low cytotoxicity in the dark [102]. Recently, G4s served as targets for photopharmacological strategies for the first time. In this regard, a dithienylethene (DTE) ligand demonstrated selectivity for telomeric G4 with a cytotoxic activity modulated according to its photoisomeric state [63].

The identification of compounds that selectively bind to *c-MYC* G4 has been a priority, as *c-MYC* was long considered undruggable. An ellipticine analog, GQC-05, was the first-in-class *c-MYC* selective ligand. GQC-05 alters protein binding to the NHE III_1_ region within its promoter and decreases *c-MYC* mRNA, in agreement with a G4 stabilizing action [66]. In fact, GQC-05 synergizes with Navitoclax to induce cytotoxicity [144]. Following that, several ligands were rationally designed with the aim of *c-MYC* G4-specific recognition and downregulation. Among them, the bis-triazolyl carbazole ligand BTC-f [49], the cell penetrating thiazole peptide TH3 [109], the four-leaf clover-like ligand known as IZCZ-3 [73] and the difluoro-substituted quinoxaline QN-1 [96] have been described. Hybrid molecules with dual DNA-binding components (cIKP-PIP) were also designed for reading out the adjacent local duplex DNA sequence [53]. In addition, BZT-Indolium is a fluorescent and photostable probe that is highly specific for *c-MYC* G4. BZT-Indolium downregulates *c-MYC* transcription and can be used for in vitro staining and live cell imaging [50]. Recent strategies to identify new *c-MYC* G4 ligands include high-throughput screenings. For instance, in a microarray screen of 20,000 small molecules using fluorescently labeled *c-MYC* G4, a novel *c-MYC* G4-binding benzofuran scaffold compound was successfully identified [44]. Through affinity selection–mass spectrometry, a library of 50,000 compounds that were previously shown as inhibitors of *c-MYC* transcription was screened. The results demonstrated that only one ligand (compound 3) functions through *c-MYC* G4 binding [145]. Other strategies use gold-coated magnetic nanoparticles to assemble *c-MYC* G4. These nanotemplates facilitate the regioselective formation of triazole products such as Tz1, a high-affinity *c-MYC* G4 ligand [117].

Exploring the impact of *k-RAS* ligands on *RAS*-driven tumors also gained immediate interest. Since TMPyP4 and many of its derivatives suffer from inadequate selectivity, new porphyrin derivatives were designed to discriminate between different G4 topologies, with a preference for the parallel over the antiparallel conformation. In particular, porphyrin-1 (cobalt containing) and porphyrin-2 (palladium containing) [54], were shown to demonstrate high affinity towards *k-RAS* promoter G4. Furthermore, a porphyrin-based photosensitizer showed preferential binding to the 3′-end of *k-RAS* G4 [94]. The fluorescent light-up acridine orange C8 also binds and stabilizes *k-RAS* G4 with modest specificity over duplex DNA [57]. A new chemotype, compound 19, displayed a high affinity for *k-RAS* G4, with a 1:1 stoichiometry ratio and a remarkable selectivity against duplex DNA. It was also shown to inhibit the transcription of the *k-RAS* driver oncogene which had been long considered undruggable [58]. 

Increasing interest has also been directed towards *VEGF* G4 ligands as antimetastatic therapies. Se2SAP, a core-modified expanded porphyrin analogue with significantly reduced photoreactivity and increased effectivity in G4-binding as compared with TMPyP4, was the first selective *VEGF* G4 binding compound identified [103]. PM2, a perylene derivative, was demonstrated to be an effective antiangiogenic agent preferentially inducing *VEGF* intramolecular G4 formation [93]. Lastly, an oligonucleotide named as VEGFq was tested. It contains the 36 nt G-rich sequence capable of forming G4 in the *VEGF* promoter. VEGFq binds specifically to the C-rich strand, via strand invasion, stabilizing the G4 structure formed by the genomic G-rich sequence, resulting in *VEGF* transcriptional inhibition [118].

Other G4 ligands targeting “unique” oncogenic promoters have also emerged. An example is carbazole TO, a fluorescent probe that preferentially targets *BCL2* G4, whose fluorescence intensity is greatly enhanced in the presence of *BCL2* G4 [51]. Moreover, two furopyridazinone derivatives effectively bind to *BCL2* G4 with a good selectivity and inhibit *BCL2* gene transcription [65]. Topotecan was found to display a high binding affinity to *c-MYB* in vitro and effectively affects its transcription [116]. Isoalloxazines show selective binding to *c-KIT* G4 and provide a proof of concept for the inhibition of *c-KIT* transcription [72]. A small drug-like pharmacological chaperone molecule, GTC365, partially directs the correct *hTERT* G4 folding pathway, reducing *hTERT* activity through its transcriptional repression [68]. The ellipticine analog, GSA1129, selectively targets the 3′-end *PDGFRb* G4, to favor its structure and downregulates the transcription of this growth factor receptor [67]. The natural alkaloid liensinine demonstrates high affinity and selectivity for the G4 formed in the promoter of the growth factor receptor *FGFR2*, inhibiting its activity [75]. Furthermore, cepharanthine, a nonplanar molecule derived from the Chinese herb *Stephania cepharantha*, was found to recognize and stabilize the G4 in the 3′-flanking region of the signal transducer *STAT3*, downregulating its transcription [52].

In contrast, other novel ligands were reported to be selective compounds interacting not only with a unique G4, but also with a narrow-spectrum of G4s. Among them, new substituted diquinolinyl-pyridine ligands show a preference for the parallel conformation of telomeric, *c-MYC* and *c-KIT* G4s [59]. The same selectivity pattern was confirmed for APTO-253, a phenanthroline derivative that is in phase I clinical trials for the treatment of acute myeloid leukemia [41]. S4-5, a furobenzoxazine naphthoquinone, behaves as a multi-target ligand with ability to bind to both telomeric and *c-MYC* extratelomeric structures with promising biological results [106]. The triarylpyridine 20A is another example of ligand shown to affect multiple G4s including telomeric, *c-KIT* and *k-RAS* [120]. Lastly, a prolinamide-derived peptidomimetic molecule triggers cell death by synthetic lethality thanks to the simultaneous inhibition of the transcription of *c-MYC* and *BCL2* genes through their respective promoter G4s [95]. 

### 2.2. RNA G4 Ligands

As described in the Background Section, increasing evidence suggests that UTRs, coding sequences, and splicing sites of cancer-relevant genes contain putative RNA G4s, which can be targeted. In addition, RNA G4s play important roles associated with telomeric function. For example, TMPyP4 binds to a *TERRA* G4 dimer, intercalating into the 5′-5′ stacking interface of two G4s blocks with a binding stoichiometry of 1:1 [146]. Hereafter, we compile the published information regarding compounds identified as RNA G4 ligands targeting 5′UTR, splicing sites and miRNAs of relevance in cancer (Table 2).

#### 2.2.1. 5′UTR G4 Ligands

The discovery of RNA G4s in the 5′UTRs of numerous genes led to the proposal that such RNA motifs could be suitable targets for small molecules to modulate mRNA translation. The two quinolinium derivatives, RR82 and RR110, considerably reduced the translational efficiency of the *n-RAS* 5′UTR [162]. After them, several compounds with this ability were discovered. Both a polyaromatic molecule, RGB-1 [161], and a novel p-(methylthio)styryl-substituted quindoline derivative [160], stabilize *n-RAS* G4 and subsequently repress *n-RAS* translation. An anionic phthalocyanine, ZnAPC, is another G4 ligand that even binds to *n-RAS* in the presence of abundant RNA in mammalian cells, resulting in selective cleavage of the targeted G4 upon photo-irradiation [167]. In addition to *n-RAS*, its *k-RAS* counterpart is also targeted by specific G4 ligands. In this regard, the alkyl derivative of TMPyP4 (TMPyP4-C14), binds to G4s in the 5′UTR of *k-RAS* mRNA and, upon photoactivation, selectively induces mRNA degradation, resulting in *k-RAS* protein downregulation by approximately 90% [166]. A biotin-streptavidin pull-down assay identified an anthrafurandione as a potent binder for G4s in the 5′UTR of *k-RAS* transcript, repressing its translation in a dose-dependent manner [148]. In a multitarget strategy attempt, alkyl cationic porphyrins were reported to penetrate the cell membrane and bind to *k-RAS* and *n-RAS* mRNAs, while generating reactive oxygen species upon photoirradiation and finally downregulating both *k-RAS* and *n-RAS* expression. Therefore, these alkyl porphyrins are efficient photosensitizers for the photodynamic therapy of *RAS*-driven cancers [147]. Beyond *RAS*, G4 targeting of other cancer-relevant genes is increasingly required. Three bisquinolinium compounds (360A, Phen-DC3 and Phen-DC6) selectively bind to the telomere shelterin protein *TRF2* G4, inhibiting its protein expression in a cellular context [151]. A quinazoline derivative exhibits a significant and specific interaction with the G4 in *VEGFa* 5′UTR, downregulating *VEGFa* translation and significantly impeding tumoral cell migration [159]. A series of new methylquinolinium derivatives have been synthesized and among them, C-24 showed selective affinity for the G4 harbored in the metalloproteinase *ADAM10* 5′UTR, strongly upregulating its translation [153]. Moreover, pyridostatin specifically potentiates the translational suppressing effect of the G4 located at 5′UTR of the nuclear factor *HNF4a* [158]. A G4 located at the 5’end of the antiapoptotic cochaperone *BAG1* 5’UTR is stabilized by small-molecule ligands such as carboxypyridostatin and Phen-DC3, which reduce the expression of endogenous *BAG1* isoforms [149]. 

#### 2.2.2. Splicing Site G4 Ligands

G4s are also present at RNA splicing sites, which reveals a novel and significant role in regulating alternative splicing and expression patterns, and thus, makes them a possible target for antitumoral G4 ligands. For example, the ellipticine GQC-05, antagonizes the major 5’ splicing site of *BCLX* gene, inhibiting the expression of its antiapoptotic isoform, activating the alternative 5’ splicing site, and expressing a proapoptotic isoform. Apparently, these effects are due to specific interactions between GQC-05 and G4s on *BCLX* pre-mRNA [154]. The aforementioned small molecule, pyridostatin, also binds to G4s in the pre-mRNA of the *EWSR1* protein involved in sarcoma translocations, blocking its interaction with the RNA-binding protein HNRNPH1 and regulating its splicing [157]. Similarly, C-12459 induces an alteration of the *hTERT* splicing pattern, causing the almost complete disappearance of the active transcript and an overexpression of the inactive transcript [31]. In addition, a high-throughput screen identified emetine and its analog, cephaeline, as small molecules that disrupt G4s, resulting in the inhibition of G4-dependent alternative splicing in a genome-wide manner [152].

#### 2.2.3. miRNA G4 Ligands

Recent efforts have focused on the development of ligands targeting miRNAs associated with cancer pathways, as they would allow for the simultaneous regulation of multiple mRNAs involved in carcinogenesis. Various well-known G4 binders were reported to target several miRNAs of relevance in cancer. In this way, TMPyP4 stabilizes G4s in *miRNA-1587* [165] and *miRNA-149* [164] and Phen-DC3 interacts with *pre-miRNA-149* [150]. Moreover, two synthesized jatrorrhizine derivatives with terminal amine groups induce the dimerization of *miRNA-1587* G4 forming 1:1 and 2:1 complexes with the dimeric G4, although the derived effect has not yet been studied [155]. Interestingly for *TP53*-driven tumors, sanguinarine was revealed as a high affinity binder with stabilization effects on the *miRNA-3620-5p* G4, blocking the base-pairing of *miRNA-3620-5p* with its target sequence [163]. Moreover, a strategy based on rationally designed locked nucleic acid (LNA) emerged for miRNA targeting. An LNA was designed to bind specifically to the G4 conformation of *pre-miRNA-92b* and was shown to inhibit its maturation. Consequently, LNA treatment rescues *PTEN* expression, which is suppressed by the elevated level of *miRNA-92b* in cancer [156].

### 2.3. I-Motif Ligands

While there are hundreds of ligands that interact with DNA and RNA G4s, there are very few compounds that target i-Motifs. Several ligands act as dual i-Motif/G4-interactive compounds. Thus, we would like to highlight the importance of evaluating their behavior towards the i-Motif counterpart, when studying G4-targeting compounds. In this section, DNA i-Motif ligands that were reported to exert antitumoral activity are grouped based on telomeric or extratelomeric targeting (Table 3).

#### 2.3.1. Telomeric i-Motif Ligands

The first study of a small molecule binding to an i-Motif was reported for TMPyP4, which promotes the formation of the telomeric i-Motif. This study raised the intriguing possibility that TMPyP4 triggers the formation of non-B DNA structures in both strands of the telomeres through a nonintercalative mechanism [179]. The drug mitoxantrone and some analogues bind to telomeric and *c-MYC* i-Motif forming sequences, even at physiological pHs [181]. The interaction of well-known G4 ligands, such as berberine, BRACO-19, mitoxantrone, Phen-DC3, pyridostatin and RHPS4, with the i-Motif-forming sequence on telomeric DNA has also been confirmed [168]. Apart from these dual telomeric G4 and i-Motif binders, various compounds have been revealed to interact exclusively with i-Motif in telomeres. Single-walled carbon nanotubes (SWNTs) were the first nanodevice capable of inhibiting DNA duplex association and selectively, thus inducing human telomeric i-Motif formation by binding to the 5-end major groove [178,182]. In addition, phenanthroline derivatives can stabilize the structure of the human telomeric i-Motif [177]. The cyclic tetraoxazole compound (L2H2-4OTD) is the most recent compound to be characterized as a telomeric i-Motif ligand with two molecules binding cooperatively [175]. 

#### 2.3.2. Extratelomeric i-Motif Ligands

The model ligand TMPyP4 also targets the i-Motif found in the chromatin remodeler *SMARCA4,* causing its destabilization [180]. To date, several *BCL2* i-Motif ligands have been discovered. The first one was a cholestane derivative, compound NSC138948 (IMC-48). IMC-48 traps out the *BCL2* i-Motif, shifting the equilibrium of secondary DNA structures and causing transcriptional overexpression of *BCL2* [173]. Additionally, three natural flavonoids (P1, P5 and P6) exhibit a clear affinity for *BCL2* i-Motif binding at a 1:1 stoichiometry [172]. Enantioselectivity is also important for i-Motif targeting as is the case for the peptidomimetic ligands PBP1 and PBP2. The para-isomer PBP1 exhibits high selectivity for the *BCL2* i-Motif whilst the meta-isomer PBP2 selectively binds to *BCL2* G4. As a consequence, PBP1 upregulates and PBP2 downregulates *BCL2* gene expression in cancer cells [88]. Other oncogenic i-Motifs have gained attention for targeting. In the same study in which the molecule GSA11129 was identified, a benzothiophene-2-carboxamide (NSC309874) demonstrated *PDGFRb* i-Motif-interactive selectivity, inhibiting *PDGFRb* promoter activity [67]. In this line, nitidine, a benzophenanthridine alkaloid, dissipates the hairpin species and stabilizes the *k-RAS* i-Motif. However, nitidine also stabilizes the three existing *k-RAS* G4s and this combined effect leads to the downregulation of *k-RAS* gene expression [176]. In contrast, the acridone derivative B19 selectively stabilizes the *c-MYC* promoter i-Motif without significant binding to the respective G4 and duplex DNA, causing the downregulation of *c-MYC* transcription [169]. Interestingly, the plant flavonol, fisetin, preferentially binds to the i-Motif from the *VEGF* promoter region with theranostic applicability. Fisetin does not induce the stabilization of the *VEGF* i-Motif structure but causes both fluorescence emission and the transformation of the i-Motif into a hairpin-like structure; thus, it can be used to diagnose aberrant formations in i-Motifs. Furthermore, fisetin facilitates the processivity of polymerases and this control of replication by fisetin is therapeutically important and feasible [171]. Very recently, different i-Motif nanotemplates, such as gold-coated magnetic nanoparticles functionalized with the *c-MYC* and *BCL2* i-Motifs, were employed to promote the metal-free synthesis of specific i-Motif ligands. In order to generate selective ligands for i-Motifs over G4s and duplex DNAs, complementary *c-MYC* and *BCL2* G4s and self-complementary duplex DNA functionalized nanotemplates have been used as control templates. Such a strategy generated cell-membrane permeable triazole leads. In vitro studies reveal that the *c-MYC* i-Motif leads to the downregulation of *c-MYC* gene expression whereas the *BCL2* i-Motif leads to the upregulation of *BCL2* gene expression [174]. Thus far, only one compound is considered to be a pan-i-Motif ligand: C343, a coumarin derivative that exhibits its selectivity for i-Motif DNA over G4s and duplexes due to its unique recognition based on hemi-protonated C-bases with negatively charged functionality. Unlike other previously reported i-Motif ligands, C343 stands out due to its detection versatility. It can sense various i-Motifs with different chain lengths, sizes, molecularities, and loop lengths, including both intramolecular and intermolecular structures [170].

## 3. Discussion

Cancer is a major disease that poses a serious threat to human life and health. As a result of its complex and heterogeneous pathogenesis, there are still many challenges in cancer therapy. Finding novel anti-tumor drugs with high selectivity and few side effects is still the main focus of cancer research. As demonstrated in the present review, an imbalance in quadruplex dynamics contributes to carcinogenesis, and its manipulation by quadruplex ligands provides a novel opportunity to defeat cancer. Initial efforts were mainly focused on targeting telomeric quadruplexes in order to inhibit telomere extension in cancer cells using telomerase, whereas later studies attempted to transcriptionally modulate individual cancer genes by targeting their quadruplexes. Although there is a long way to go in the development of potent drugs, various promising lead compounds have been obtained; however, the results have thus far been limited. Firstly, at this stage, the variety of binding sites for these ligands and the differences in their effects on the quadruplex structures make it difficult to unravel how quadruplexes influence biological function, i.e., whether the stabilization or destabilization of quadruplexes promotes or inhibits gene expression. Secondly, the correlation between stabilization in vitro and cell activity is not straightforward. In particular, a G4 target characterized in vitro may not be the sole G4 targeted in cells. Furthermore, there is also inherent cell variability, which has an impact on the relationship between in vitro and in vivo results. A further point to be addressed for the majority of ligands described thus far is that they are generally characterized by high-molecular weights and protonated side chains, which may affect their cellular uptake. However, the major limitation for the clinical application of quadruplex ligands seems to be directly related to selectivity. In fact, the selectivity pattern of several quadruplex ligands is dose-dependent. Although global or multiple G4 targeting approaches may be effective, targets need to be clearly defined in advance. Other conceivable obstacles are the potential side effects of the ligands on normal tissues. Moreover, the predictive response biomarkers need to be identified if a personalized anticancer management is to be achieved. Nevertheless, given the rapid accumulation of data on quadruplex structures and the related biological functions, and the rapid development of ligands, we are confident that these limitations can be overcome. In this regard, a wealth of new derivatives with lower cytotoxicity and superior selectivity will emerge in the near future.

## 4. Conclusions

In this review, we give an overview of a range of compounds that target quadruplexes, including DNA G4s, RNA G4s, and i-Motifs, and discuss their limitations. The quadruplex-mediated antitumoral effects reported herein may pave the way for cutting-edge therapeutic approaches in the future treatment of human cancer. 

## Figures and Tables

**Figure 1 cancers-13-03156-f001:**
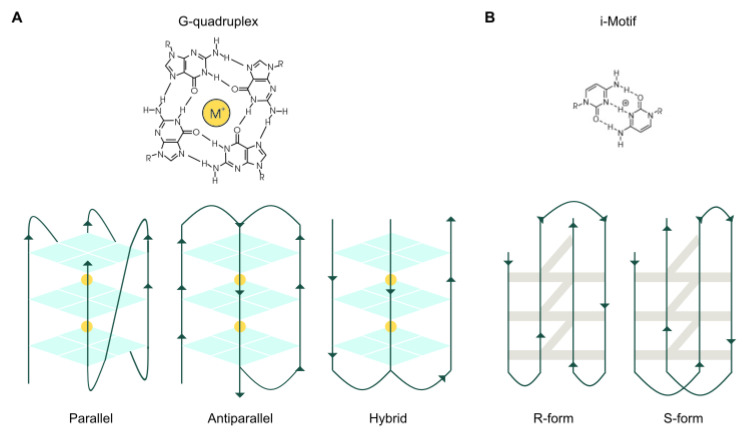
Quadruplex structures. (**A**) Chemical structure of a G4 and schematic representation of various G4 topologies. (**B**) Chemical structure of an i-Motif and schematic representation of various i-Motif topologies.

**Figure 2 cancers-13-03156-f002:**
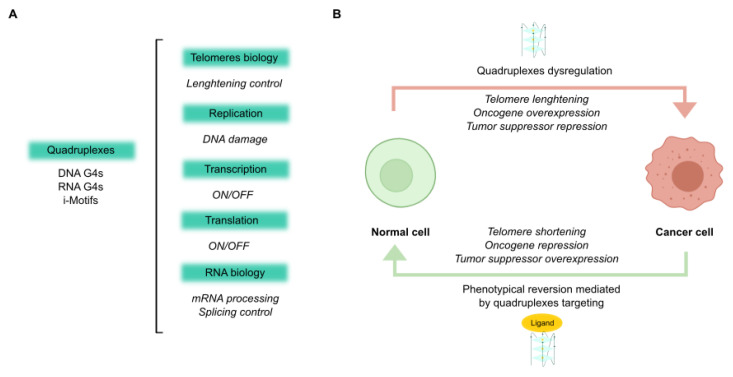
Quadruplex functions. (**A**) Schematic diagram of the multiple roles of quadruplexes in several cellular events. (**B**) Representation of antitumoral effects mediated by quadruplex ligands for cancer therapy.

**Figure 3 cancers-13-03156-f003:**
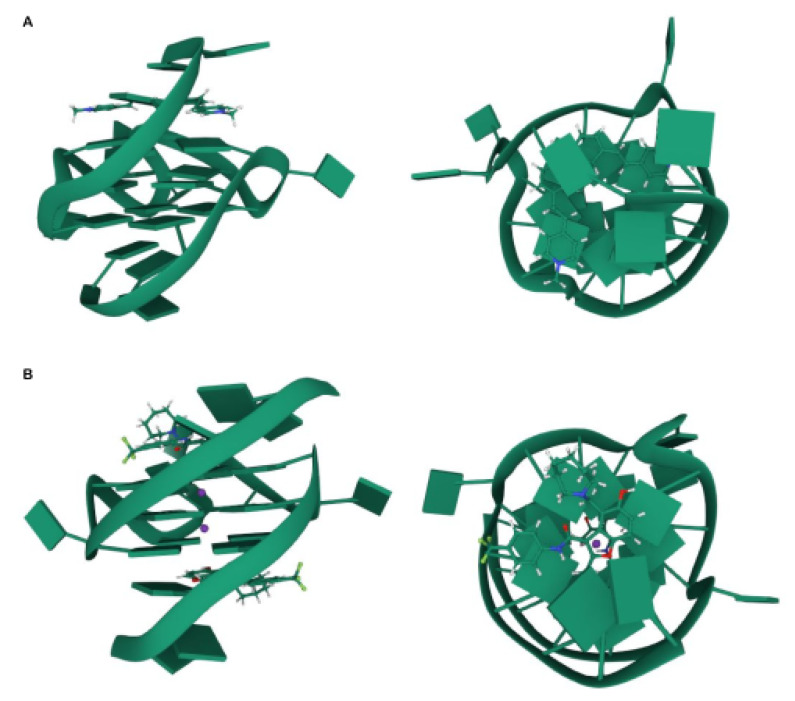
Quadruplex ligands with different binding modes. (**A**) NMR structure from different rotation angles of human *C-MYC* G4 (TGAGGGTGGGTAGGGTGGGTAA) bound to a carbazole derivative (Protein Data Bank: 6JJ0). (**B**) NMR structure from different rotation angles of the same human *C-MYC* G4 than (**A**) bound to a benzofuran-containing compound (Protein Data Bank: 5W77).

**Figure 4 cancers-13-03156-f004:**
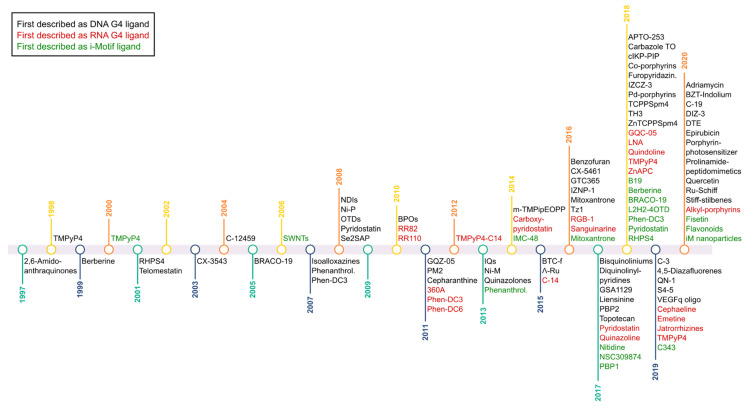
Timeline for quadruplex interactive ligands. Timeline showing the year in which the small-molecule compounds were first described as quadruplex interactive ligands. DNA G4 ligands are in black, RNA G4 ligands are in red and i-Motif ligands are in green.

**Figure 5 cancers-13-03156-f005:**
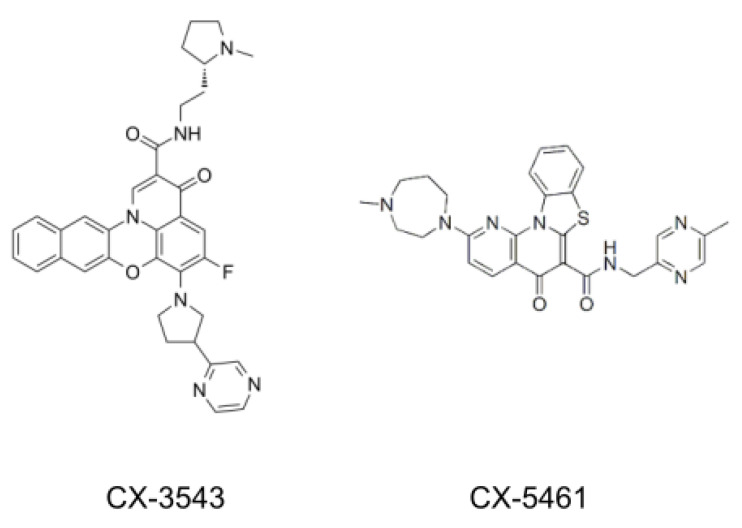
Chemical structure of quadruplex ligands used in clinical trials. CX-3543 and CX-5461 are included.

**Figure 6 cancers-13-03156-f006:**
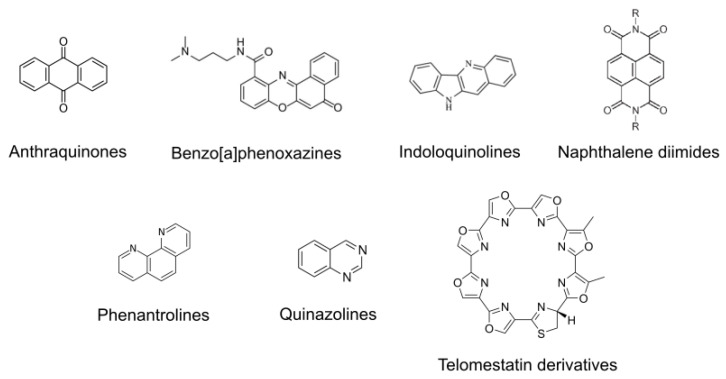
Chemical structure of the main chemo-families of G4 ligands. Anthraquinones, Benzo[a]phenoxazines, Indoloquinolines, Naphtahlene diimides, Phenantrolines, Quinazolines and Telomestatin derivatives are shown.

**Table 1 cancers-13-03156-t001:** DNA G4-interacting ligands reported to exhibit antitumoral effect and their targets. Ligands that have been demonstrated to possess antitumoral activity in vivo are marked with an asterisk.

Ligand	Target	Effect
Adriamycin *	*Tel* [40]	Telomere dysfunction
APTO-253 *	*c-KIT* [41]; *c-MYC* [41]; *Tel* [41]	Inhibition of oncogene transcription and telomere dysfunction
Benzo[a]phenoxazines	*c-KIT* [42]; *Tel* [43]	Inhibition of oncogene transcription and telomere dysfunction
Benzofuran	*c-MYC* [44]	Inhibition of oncogene transcription
Berberine *	*c-KIT* [45]; *k-RAS* [46]; *Tel* [47]	Inhibition of oncogene transcription and telomere dysfunction
BRACO-19 *	*Tel* [48]	Telomere dysfunction
BTC-f	*Tel* [49]	Telomere dysfunction
BZT-Indolium	*c-MYC* [50]	Inhibition of oncogene transcription
Carbazole TO	*BCL2* [51]	Inhibition of oncogene transcription
Cepharanthine	*STAT3* [52]	Inhibition of oncogene transcription
cIKP-PIP	*c-MYC* [53]	Inhibition of oncogene transcription
Co- and Pd-porphyrins *	*k-RAS* [54]	Inhibition of oncogene transcription
CX-3543 *	*Pan-binder* [55]	Inhibition of oncogene transcription
CX-5461 *	*Pan-binder* [55]	Inhibition of oncogene transcription
C-3	*c-MYC* [56]	Inhibition of oncogene transcription
C8	*k-RAS* [57]	Inhibition of oncogene transcription
C-19	*k-RAS* [58]	Inhibition of oncogene transcription
Diquinolinyl-pyridines	*c-KIT* [59]; *c-MYC* [60] *Tel* [59]	Inhibition of oncogene transcription and telomere dysfunction
Disubstituted amidoanthraquinones	*Tel* [61]	Telomere dysfunction
DIZ-3	*Tel* [62]	Telomere dysfunction
DTE	*Tel* [63]	Telomere dysfunction
Epirubicin *	*Tel* [64]	Telomere dysfunction
Furopyridazinones	*BCL2* [65]	Inhibition of oncogene transcription
GQC-05	*c-MYC* [66]	Inhibition of oncogene transcription
GSA1129	*PDGFRb* [67]	Inhibition of oncogene transcription
dGTC365	*hTERT* [68]	Telomere dysfunction
Indoloquinolines *	*c-KIT* [69]; *c-MYC* [70]; *k-RAS* [69]; *Tel* [69]; *VEGF* [71]	Inhibition of oncogene transcription and telomere dysfunction
Isoalloxazines	*c-KIT* [72]	Inhibition of oncogene transcription
IZCZ-3 *	*c-MYC* [73]	Inhibition of oncogene transcription
IZNP-1	*Tel* [74]	Telomere dysfunction
Liensinine *	*FGFR2* [75]	Inhibition of oncogene transcription
Mitoxantrone *	*Tel* [76]; *WT1* [77]	Inhibition of oncogene transcription and telomere dysfunction
m-TMPipEOPP	*Tel* [78]	Telomere dysfunction
Naphthalene diimides *	*BCL2* [79]; *c-KIT* [80]; *MDM2* [81]; *Ribosomal DNA* [82]; *Tel* [83]	Inhibition of oncogene transcription and telomere dysfunction
Ni-M	*Tel* [84]	Telomere dysfunction
Ni-P	*Tel* [85]	Telomere dysfunction
Nitidine *	*k-RAS* [73]	Inhibition of oncogene transcription
Oxazole telomestatins	*Tel* [86,87]	Telomere dysfunction
PBP2	*BCL2* [88]	Inhibition of oncogene transcription
Phenanthrolines	*c-KIT* [89]; *c-MYC* [89]; *Tel* [90]	Inhibition of oncogene transcription and telomere dysfunction
Phen-DC3	*c-MYC* [91]; *Tel* [92]	Inhibition of oncogene transcription and telomere dysfunction
PM2	*VEGF* [93]	Inhibition of oncogene transcription
Porphyrin- photosensitizer	*k-RAS* [94]	Inhibition of oncogene transcription
Prolinamide- peptidomimetic	*BCL2* [95]; *c-MYC* [95]	Inhibition of oncogene transcription
Pyridostatin *	*Pan-binder* [19]	Inhibition of oncogene transcription
QN-1	*c-MYC* [96]	Inhibition of oncogene transcription
Quercetin *	*Tel* [97]	Telomere dysfunction
Quinazolines	*c-KIT* [98]; *c-MYC* [99]; *RET* [100]	Inhibition of oncogene transcription
RHPS4 *	*Tel* [101]	Telomere dysfunction
Ru-Schiff	*Tel* [102]	Telomere dysfunction
Se2SAP	*VEGF* [103]	Inhibition of oncogene transcription
Schizocommunins *	*Tel* [104]	Telomere dysfunction
Stiff-stilbenes	*Tel* [105]	Telomere dysfunction
S4-5	*c-MYC* [106]; *Tel* [106]	Inhibition of oncogene transcription and telomere dysfunction
Telomestatin *	*c-MYB* [107]; *Tel* [108]	Inhibition of oncogene transcription and telomere dysfunction
TH3	*c-MYC* [109]	Inhibition of oncogene transcription
TMPyP4 *	*BCL2* [110]; *c-MYC* [111]; *k-RAS* [112]; *PDGFa* [113]; *Tel* [114]; *VEGF* [115]	Inhibition of oncogene transcription and telomere dysfunction
Topotecan *	*c-MYB* [116]	Inhibition of oncogene transcription
Tz1	*c-MYC* [117]	Inhibition of oncogene transcription
VEGFq oligo	*VEGF* [118]	Inhibition of oncogene transcription
(Zn)TCPPSpm4	*Tel* [119]	Telomere dysfunction
20A *	*c-KIT* [120], *k-RAS* [120], *Tel* [120]	Inhibition of oncogene transcription and telomere dysfunction
4,5-diazafluorenes	*Tel* [121]	Telomere dysfunction
Λ-Ru	*Tel* [122]	Telomere dysfunction

**Table 2 cancers-13-03156-t002:** RNA G4-interacting ligands reported to exhibit antitumoral effect and their targets. Ligands that have been demonstrated to possess antitumoral activity in vivo are marked with an asterisk.

Ligand	Target	Effect
Alkyl porphyrins *	*k-RAS* [147]; *n-RAS* [147]	Inhibition of oncogene translation
Anthrafurandione	*k-RAS* [148]	Inhibition of oncogene translation
Bisquinoliniums	*BAG1* [149]; *pre-miRNA149* [150]; *TRF2* [151]	Inhibition of oncogene translation and alteration of miRNA biogenesis
Carboxypyridostatin	*BAG1* [149]	Inhibition of oncogene translation
Cephaeline	*Pan-binder* [152]	Inhibition of oncogene translation
C-12459	*hTERT* [31]	Telomere dysfunction
C-14	*ADAM10* [153]	Inhibition of oncogene translation
Emetine *	*Pan-binder* [152]	Inhibition of oncogene translation
GQC-05	*BCLX* [154]	Inhibition of oncogene translation
Jatrorrhizines	*miRNA-1587* [155]	Alteration of miRNA targeting
LNA	*pre-miRNA-92b* [156]	Alteration of miRNA biogenesis
Pyridostatin *	*EWSR1* [157]; *HNF4a* [158]	Inhibition of oncogene translation
Quinazolines *	*VEGF* [159]	Inhibition of oncogene translation
Quindolines	*n-RAS* [160]	Inhibition of oncogene translation
RGB-1	*n-RAS* [161]	Inhibition of oncogene translation
RR82 and RR110	*n-RAS* [162]	Inhibition of oncogene translation
Sanguinarine *	*miRNA-3620-5p* [163]	Alteration of miRNA targeting
TMPyP4 *	*miRNA-149* [164]; *miRNA-1587* [165]; *TERRA* [146]	Alteration of miRNA targeting and telomere dysfunction
TMPyP4-C14	*k-RAS* [166]	Inhibition of oncogene translation
ZnAPC	*n-RAS* [167]	Inhibition of oncogene translation

**Table 3 cancers-13-03156-t003:** I-Motif-interacting ligands reported to exhibit antitumoral effect and their targets. Ligands that have been demonstrated to possess antitumoral activity in vivo are marked with an asterisk.

Ligand	Target	Effect
Berberine *	*Tel* [168]	Telomere dysfunction
BRACO-19 *	*Tel* [168]	Telomere dysfunction
B19	*c-MYC* [169]	Inhibition of oncogene transcription
C343	*Pan-binder* [170]	Inhibition of oncogene transcription
Fisetin	*VEGF* [171]	Inhibition of oncogene transcription
Flavonoids	*BCL2* [172]	Inhibition of oncogene transcription
IMC-48 *	*BCL2* [173]	Inhibition of oncogene transcription
iM nanoparticles	*BCL2* [174]; *c-MYC* [174]	Inhibition of oncogene transcription
L2H2-4OTD	*Tel* [175]	Telomere dysfunction
Mitoxantrone *	*Tel* [168]	Telomere dysfunction
Nitidine *	*k-RAS* [176]	Inhibition of oncogene transcription
NSC309874	*PDGFRb* [67]	Inhibition of oncogene transcription
PBP1	*BCL2* [88]	Inhibition of oncogene transcription
Phenanthrolines	*Tel* [177]	Telomere dysfunction
Phen-DC3	*Tel* [168]	Telomere dysfunction
Pyridostatin *	*Tel* [168]	Telomere dysfunction
RHPS4 *	*Tel* [168]	Telomere dysfunction
SWNTs *	*Tel* [178]	Telomere dysfunction
TMPyP4 *	*Tel* [179]; *SMARCA4* [180]	Inhibition of oncogene transcription and telomere dysfunction

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
