# Peer review of "Quadruplex Ligands in Cancer Therapy"

_cancers, 2021, doi:10.3390/cancers13133156_

Round 1

Reviewer 1 Report

This review article aims to “concisely review the latest progress made in quadruplexes targeting in antitumoral therapy”. The area has a very large number of reviews – nearly 600 are listed in PUBMED, and 54 in 2021. So, any new review, if it is to be useful, has to offer some new insights and be of real value to the community.

This review has some useful things to say but needs revision on several counts:

  1. Its brief review of quadruplexes is insubstantial and there is no mention of the NMR and crystallographic studies. Figure 1 is unclear and needs redrawing. Potassium ions, in particular, are found midway between successive G-quartets, not as the figure implies, within a quartet.
  2. The table of “antitumoral effects”, Table 1, needs to distinguish between those compounds for which antiproliferative activity in cancer cell lines has been reported, and those for which in vivo anticancer activity has been demonstrated. The two categories are not the same at all.
  3. References are needed for the hTERT section on page 3.
  4. Figure 3 shows an intermolecular quadruplex and is not representative of human telomeric drug quadruplexes. It does not show the binding mode common to almost all G4 ligands. There are better and more relevant NMR and crystal structures that could be used.
  5. The web site G4LDB is not currently attainable
  6. BRACO19 was first reported in 2001 (Read et al., PNAS), not 2005 and the 2005 paper quoted here is a later report from the same group on in vivo activity. Figure 4 needs revision
  7. Figure 4 is also misleading in stating that drugs such as mitoxantrone were described for the first time at particular dates. These may well be the dates when G4 affinity was described, but many entries in the Figure have been known well before these dates.
  8. It should be made clear that TMPyP4 is not a selective ligand and its effects may well be duplex related.
  9. The review is not up to date and fails to cover some major developments in 2020, for example (a) the recent work by the Balasubramanian laboratory defining promoter sites is very relevant to anticancer action, (b) in the naphthalene diimide field with the use of whole-genome transcriptional sequencing to determine targets, the demonstration of in vivo activity for novel potent compounds and activity in a drug-resistant cell line
  10. It does not adequately mention or critically assess the issues of quadruplex/gene selectivity and off-target effects. The section on selectivity (page 4/5) does not mention that most ligands have not been evaluated in large panels of G4s or across the human genome, so selectivity is not really correctly assigned. Many of the G4 ligands discussed here are rather promiscious
  11. The reference list has several errors and needs a careful re-read (refs 49, 55, 59, 145)
  12. The text needs revision by a native English speaker.

Author Response

We greatly appreciate your comments. Please see the attachment with our point-by-point response. 

Reviewer 2 Report

The review article “Quadruplex ligands in cancer therapy” highlights the application of three different quadruplex structures in cancer therapy. The manuscript is well-structured and written. The content is ambitious and different from that it is already published. Before considering this manuscript for publication, minor comments need to be revised as follows:

Specific Comments:

*Line 59: “Experiments using chemical, molecular and cell biology methods have demonstrated that quadruplexes are very numerous in the human genome.” The authors can mention which are methods can be used and add references.

*Line 101: eliminate the word “microRNAs”

*Line 101: “multiple microRNAs”. The authors should explain in more detail.

*Table 1: Other relevant ligands should be provided, for instance, Phthalocyanines, Schizocommunin derivative, C8, Anthrafurandione derivative, 20A.

*Table 1, 2 and 3: The authors should add a column with the biological effects of each ligand.

Author Response

(The authors gave the same response as above.)
